Ecological distribution of protosteloid amoebae in New Zealand

Zahn Geoffrey gzahn@email.uark.edu
Stephenson Steven L.
Spiegel Frederick W.
Department of Biological Sciences, University of Arkansas , Fayetteville, AR , USA
Buckley Hannah
Electronic publication date: 2014 Mar 11
Publication date: 2014
Volume: 2
Electronic Location ID: e296
Received 2013 Dec 24; Accepted 2014 Feb 10
Copyright: © 2014 Zahn et al.
Copyright year: 2014
Copyright holder: Zahn et al.
License: This is an open access article distributed under the terms of the Creative Commons Attribution License, which permits unrestricted use, distribution, and reproduction in any medium, provided the original author and source are credited.
License URL: https://creativecommons.org/licenses/by/3.0/

Keywords: Amoebae, Protostelid, New Zealand, Biogeography

Funding: National Science Foundation DEB-0316284 National Geographic Society 6051-97 4732-92 This project was partially funded by the National Science Foundation (DEB-0316284) and The National Geographic Society (6051-97 and 4732-92). The funders had no role in study design, data collection and analysis, decision to publish, or preparation of the manuscript.

==============================
During the period of March 2004 to December 2007, samples of aerial litter (dead but still attached plant parts) and ground litter (dead plant material on the ground) were collected from 81 study sites representing a wide range of latitudes (34°S to 50°S) and a variety of different types of habitats throughout New Zealand (including Stewart Island and the Auckland Islands). The objective was to survey the assemblages of protosteloid amoebae present in this region of the world. Twenty-nine described species of protosteloid amoebae were recorded by making morphological identifications of protosteloid amoebae fruiting bodies on cultured substrates. Of the species observed, Protostelium mycophaga was by far the most abundant and was found in more than half of all samples. Most species were found in fewer than 10% of the samples collected. Seven abundant or common species were found to display significantly increased likelihood for detection in aerial litter or ground litter microhabitats. There was some evidence of a general correlation between environmental factors - annual precipitation, elevation, and distance from the equator (latitude) - and the abundance and richness of protosteloid amoebae. An increase in each of these three factors correlated with a decrease in both abundance and richness. This study provides a thorough survey of the protosteloid amoebae present in New Zealand and adds to a growing body of evidence which suggests several correlations between their broad distributional patterns and environmental factors.

Introduction

The term “protosteloid amoebae” refers to a paraphyletic assemblage of unicellular eukaryotes within the supergroup Amoebozoa that exhibit spore dispersal via sporocarpic fruiting (See Fig. S1). For most of their life cycle, protosteloid amoebae exist as single amoeboid cells that may or may not possess flagella (Shadwick et al., 2009). These organisms are thought to be important consumers of bacteria and other microorganisms (Adl & Gupta, 2006). Although global inventories carried out thus far suggest that protosteloid amoebae occur in every type of terrestrial system (Ndiritu, Stephenson & Spiegel, 2009), very little is known about their ecology. The results obtained from previous studies (Moore et al., 2000; Spiegel & Stephenson, 2000; Stephenson et al., 2004) have provided some evidence that ecosystems located at higher latitudes support fewer species and show a decline in species abundance. Because of its location, size, and isolation, New Zealand provided an excellent opportunity to investigate these patterns.

New Zealand is the most isolated land mass of its size in the world (Cavender et al., 2002) and represents a unique collection of ecosystems with highly endemic flora (Fleet, 1986). Protosteloid amoebae have been known from New Zealand (Olive & Stoianovitch, 1969), and is the location from which the type specimen of Schizoplasmodium cavostelioides was originally isolated (Olive, 1967). The primary focus of the present study was to exhaustively sample as much of this range as possible in order to characterize the ecological distribution of the protosteloid amoebae present.

Materials and Methods

During the period of March 2004 to December 2007, three separate collecting trips were made to 81 sites on the North Island (113,729 km2), South Island (151,215 km2) and the Auckland Islands (625 km2) (Fig. 1 and Table S1). Samples were obtained from Stewart Island (1,746 km2) in 2006, but yielded no observations of protosteloid amoebae. Collectively, the study sites sampled represent a well-characterized and diverse array of habitats encompassing a variety of elevations (extending from 0 m to 1636 m), every major vegetation type found in New Zealand, and a rather wide range of latitudes, from 34.44°S to 50.85°S. A total of 247 samples of aerial litter and 234 samples of ground litter were taken collected from 81 different study sites. These samples were placed in small paper bags, air dried, and transported to the laboratory for processing. In order to achieve a broad coverage of many different types of dead plant material (substrates), sampling efforts did not include systematic replications of substrate types or habitats, but multiple samples from many habitats were collected. Ecosystem types ranged from beaches and open roadsides to tree fern forests and alpine tundra (see Table S1).

Figure 1 Map of sampling locations.

Sample site markers are scaled to represent the mean number of protosteloid amoebae fruiting bodies encountered for each line of substrate observed from that site. N = species richness observed at each major latitudinal range.

In the laboratory, within 3 months of collection, samples were cut into small pieces, wetted with sterile water, and plated in lines on minimal nutrient agar (0.002 g malt extract, 0.002 g yeast extract, 0.75 g K2HPO4, 15.0 g Difco Bacto Agar, 1.0 L deionized [DI] H2O) as described by Spiegel et al. (2004), yielding 6,533 lines of substrate that were examined in 1,175 plates. Lines of substrate consisted of approximately 2 cm × 0.5 cm wetted strips of dead plant matter gently pressed to the surface of the agar (see Fig. S2). Daily observations were made for a minimum of seven days using bright-field microscopy with the 10X objective lens on a Zeiss Axioskop 2 microscope. Species were identified based on sporocarp morphology according to Olive (1967), Olive (1970) and Spiegel et al. (2010). Observations of amoeboid and prespore stages were carried out to corroborate sporocarp identifications when necessary. This method provides a quick way to assess presence/absence of these amoebae since sporocarps are easy to detect and morphologically distinct from each other.

Species observations were recorded as presence or absence for each plated line of substrate and this resolution was used for comparisons between sites. Since sites were surveyed with varying numbers of lines of substrate, abundance and richness data were scaled by dividing by the total number of lines from a specific sample to represent abundance and richness per line of substrate observed. Precipitation data were extracted from the New Zealand National Climate Database (http://cliflo.niwa.co.nz/) and consisted of absolute precipitation amounts from the nearest weather station in the year samples were taken. A sample-based rarefaction curve (Fig. 2) was generated using Ecosim 7 (Gotelli & Entsminger, 2009). Since data were not normally distributed, the individual effects of latitude, elevation, and precipitation gradients, and microhabitat (aerial vs. ground litter) on scaled species richness and abundance were tested with the Kruskal-Wallis test, and R2 values for linear correlations were calculated using the Pearson correlation statistic in Minitab® Statistical Software version 16.

Figure 2 Rarefaction curve of species richness and sampling effort.

Sampling effort appears sufficient to uncover the diversity of protosteloid amoebae. An increase in random sub-sampling from 200 to 300 collections only yielded an additional 2 species.

Results

Twenty-nine species of protosteloid amoebae, including the minuscule myxomycete Echinostelium bisporum, were recovered in the present study. The sample-based rarefaction curve (Fig. 2) reached a clear asymptote at this species richness. While not traditionally grouped together with the now defunct “Protostelids” (Shadwick et al., 2009), the small fruiting bodies of E. bisporum display a protosteloid growth form and are commonly encountered using the current methods, so it has been included in this study. Species were grouped into abundance categories consistent with similar studies (Aguilar, Spiegel & Lado, 2011; Ndiritu, Stephenson & Spiegel, 2009) such that species recovered from: >10% of samples = abundant; 5–10% = common; 1–5% = occasional; <1% = rare. Seven species were found to be abundant across all study site locations while ten were considered commonly occurring (Table 1). Protostelium mycophaga was by far the most commonly encountered species, accounting for twenty-five percent of all fruiting body observations. Eighty out of eighty-one sites were positive for fruiting bodies of protosteloid amoebae (99%). The only site that did not yield any observations of protosteloid amoebae, located on Stewart Island, was left out of subsequent analyses.

Table 1 Observed species.

Total species observations from all sites.

Species name	Abbreviation	Total
encounters	Frequency
per sample	Category	Aerial
encounters	Ground
encounters	
Protostelium mycophaga1 **	Pm	598	2.06	A	398	200	
Schizoplasmodiopsis pseudoendospora2 *	Sps	323	1.2	A	119	204	
Nematostelium gracile1 *	Ng	239	1.05	A	83	156	
Soliformovum irregularis 3	Si	213	1.14	A	130	83	
Schizoplasmodiopsis vulgare1 ***	Sv	197	0.95	A	40	157	
Protostelium nocturnum3 ***	Pn	182	0.98	A	136	46	
Schizoplasmodiopsis amoeboidea 4	Sa	174	1.06	A	92	82	
Protostelium arachisporum 2	Pa	73	0.33	C	43	30	
Protostelium pyriformis 1	Ppyr	57	0.41	C	27	30	
Schizoplasmodium cavostelioides 1	Sc	51	0.28	C	38	13	
Tychosporium acutostipes 5	Ta	49	0.42	C	29	20	
Cavostelium apophysatum 2	Ca	43	0.25	C	15	28	
Nematostelium ovatum 1	No	41	0.31	C	14	27	
Protostelium mycophaga 1 var. little ***	lilPm	34	0.25	C	33	1	
Endostelium zonatum 6	Ez	31	0.19	C	17	14	
Echinosteliopsis oligospora 7	Eo	28	0.2	C	14	14	
Soliformovum expulsum3 *	Se	27	0.3	C	21	6	
Echinostelium bisporum 4	Eb	16	0.16	O	7	9	
Protosteliopsis fimicola 1	Pf	12	0.12	O	7	5	
Microglomus paxillus 1	Mp	9	0.07	O	1	8	
Clastostelium recurvatum 1	Cr	8	0.09	O	3	5	
Protostelium mycophaga 1 var. repeater	Pmrep	7	0.05	O	7	0	
Schizoplasmodiopsis micropunctata 1	Sm	5	0.05	O	5	0	
Protostelium okumukumu 8	Po	5	0.05	O	1	4	
Schizoplasmodiopsis reticulata 1	Sr	4	0.01	R	2	2	
Ceratiomyxa hemisphaerica 1	Ch	2	0.01	R	0	2	
Protosporangium articulatum 1	Partic	1	0.01	R	1	0	
Protosporangium bisporum 1	Pbisp	1	0.01	R	1	0	
Schizoplasmodium obovatum 1	So	1	0.01	R	0	1	
Notes.

A abundant

C common

O occasional

R rare

* P < 0.05.

** P < 0.01.

*** P < 0.001.

(All tests: significant difference between aerial and ground litter abundance, Kruskal-Wallis test); Superscript numbers refer to naming authorities:

1 Olive and Stoianovich.

2 Olive.

3 Spiegel.

4 Olive and Whitney.

5 Spiegel, Moore, and Feldman.

6 Olive, Bennet, and Deasey.

7 Reinhardt and Olive.

8 Spiegel, Shadwick, and Hemmes.

The number of samples varied at each site due to local conditions, such as a lack of suitable standing plant material, but of the 481 total samples, 299 of them yielded identifiable fruiting bodies of protosteloid amoebae (62%). These numbers are consistent with previous studies (Aguilar, Spiegel & Lado, 2011; Ndiritu, Stephenson & Spiegel, 2009; Stephenson, Landolt & Moore, 1999). While no studies have previously examined the protosteloid amoebae of New Zealand, the methods we used for collection and observation in the previous surveys were very similar.

Microhabitat (aerial vs. ground litter) did not have a significant influence on either the abundance or species richness of fruiting amoebae as a whole (P = 0.888, Kruskal-Wallis; P = 0.746; Kruskal-Wallis, respectively), but several species displayed a significantly increased likelihood of being observed in a specific microhabitat. Of these, Protostelium mycophaga, Protostelium nocturnum, Protostelium mycophaga var. little, and Soliformovum expulsum were significantly more likely to be found on aerial litter, while Schizoplasmodiopsis pseudoendospora, Nematostelium gracile, and Schizoplasmodiopsis vulgare were more likely to be found on ground litter (Table 1). Microhabitat also made no difference to the significance of correlations between broader environmental factors (i.e., latitude, elevation, and annual precipitation) and community richness or abundance. Ecosystem type did not have any significant effect on richness or abundance, with most species displaying a cosmopolitan distribution among the different ecosystems. Species occurring in only one ecosystem type were uncommon or rare, thus it could not be determined whether these patterns were significant.

The most important factors related to protosteloid amoeba richness and abundance were elevation, precipitation and latitude (distance from the equator) (Table S2). Increases in all three factors led to perceived declines in protosteloid amoebae community measures though R2 values for linear correlations were weak (Fig. 3). The most abundant and diverse communities were typically found in drier, more northerly locations close to sea level (See Fig. 1 and Table S1).

Figure 3 Species encounters along environmental gradients.

(A–C): The scaled abundance (abundance per line of substrate observed) of protosteloid amoebae (all species). (D–F): The scaled species richness (richness per line of substrate observed). X-axis factors: Gradients of distance from equator (km, A and D), elevation (m above sea level, B and E), and annual rainfall (mm, C and F). R squared values for the linear regression are given in each panel.

Discussion

The main focus of this study was to provide a comprehensive survey of the protosteloid amoebae of New Zealand and to investigate the distribution of these species along gradients of precipitation, elevation, and latitude. A sample-based rarefaction curve (Fig. 2) suggests that sampling effort was sufficient to recover the bulk of the known and described species richness present. Broadly, we were able demonstrate that the abundance and richness of protosteloid amoebae in New Zealand were correlated with latitude, elevation, and precipitation (Table S2). However, ecosystem type did not appear to influence these relationships. Moore et al. (2000) initially suggested that latitude may play a role in the presence/absence of protosteloid amoebae when only 6 species were recovered from 80 samples in the arctic tundra. Shadwick, Stephenson & Spiegel (2009) had results more consistent with the present study, recovering 26 species from 205 samples in Great Smoky Mountains National Park, TN. In the current study microhabitat was a significant predictor of presence/absence for several species (Table 1), but the extent of this effect was far less than was reported by Aguilar, Spiegel & Lado (2011) in which only 3 out of 18 species recovered from 100 samples did not display significant differences in presence/absence between microhabitats.

The sampling method varied somewhat between collecting trips. The first and last samples collected (sampling years 2004 and 2007, Table S1) were physically separated by substrate type (i.e., a separate bag for each species of litter collected), whereas the other samples were pooled together (i.e., all aerial litter in one bag and all ground litter in another bag). This change was made for convenience, since many study sites had limited amounts of litter present and it was difficult to find substrate species that yielded both aerial and ground litter of the same species in the same general area. Cursory analysis of the two sampling methods suggested that species observations were not affected by initial pooling of samples and thus sampling methods were treated as equal for all subsequent analyses. Briefly, data from the 2004 and 2007 samples were artificially pooled within sites and randomly resampled to resemble what physically occurred in pooled sample collections. These resampled data were not significantly different from a random selection of the original unpooled data (P = 0.420, Kruskal-Wallis test). The sampling protocol did not allow for further rigorous testing of this assumption, and this is beyond the scope of the present study. Additionally, the number of plated lines of substrate per study location varied from 4 to 443 as shown in Table S1. For most sites (68%), at least forty lines of substrate were plated for observation.

These heavily observed sites may display a bias toward an increase in the observations of rare species when compared with sampling locations such as the Auckland Island sites, in which only four lines of substrate were observed. Of the five rare species identified, two (Ceratiomyxa hemisphaerica and Protosporangium bisporum) were only found at the sample location from which 443 lines were plated (Peel Forest) and none were found at any locations from which less than 32 lines were plated. These rare species account for only nine distinct observations, and excluding them from further analyses had no impact on the significance of results, so they have been left in.

The effectiveness of various levels of observational effort for the detection of protosteloid amoebae was quantified by Aguilar, Spiegel & Lado (2011) and it was found that four lines of substrate per sample was enough to detect 80% of species present, while eight lines per sample was able to yield 90% of the species present. Substantial increases in observational effort yielded only one or two additional rare species. In the present study, site richness was not significantly correlated with the number of plated lines per study location (R2 = 0.033; P = 0.103, Kruskal-Wallis test). Interestingly, six of the nine observations of rare species occurred at sites in which forty lines of substrate were plated, further suggesting that sampling efforts greater than that did little to increase the effectiveness of ecological surveys for rare species of protosteloid amoebae. It is apparent that comparisons between abundant, common, and occasional species may be safely made using the current study’s sampling and observation protocol.

This study took place over several years and samples were collected during different seasons. Though there is little evidence for true seasonality in protosteloid amoeba presence/absence (FW Spiegel, unpublished data) this must be considered when drawing conclusions from the present study. Moore & Spiegel (2000) showed that protosteloid amoebae spore dispersal was dramatically reduced in winter using artificial substrates, but on native in situ substrates, dormant stages of these amoebae persist throughout the year. Protosteloid amoebae are very tolerant of adverse conditions (drying out, etc.) and have been recovered from dried substrate at least as long as 12 years after collection (G Zahn, unpublished data) so it is likely that seasonal changes in the in situ activity of the amoebae are not reflected in the current sampling protocol, which inherently encourages encysted or dormant amoebae to reactivate and fruit. Further, in the present study, North Island sites were sampled primarily in the early austral fall and South Island sites were sampled primarily in the late austral spring. Corresponding seasons in temperate North America are excellent times to sample for protosteloid amoebae. Still, seasonal changes to substrate quality, type, and abundance are likely to have an impact on the amoebae present and may affect our results.

Supplemental Information

Figure S1 Fruiting bodies of protosteloid amoebae in situ

A cluster of sporocarps of the protosteloid amoeba Tychsporium acutostipes fruiting on a leaf. This image was taken at a total magnification of 100X. The scale bar is 100 µm. For high quality images of all species discussed in this paper, see Spiegel et al. (2007) online.

Click here for additional data file.

Table S1 Study site locations and information

Table of study sites. Habitat types are generalizations. No significant correlations between habitat type and abundance were found, either generally or by species. At some sites dead vegetation suitable as a substrate was very limited and at others it was highly abundant. Thus, the number of lines plated at each site varies from 4 to 443.

Click here for additional data file.

Figure S2 Primary isolation plate for protosteloid amoebae

A primary isolation plate with 8 lines of substrate arranged in a circle. Each line of substrate is labeled and observations of protosteloid amoebae are labeled according to which line they occurred on.

Click here for additional data file.

Table S2 Statistical test values

Kruskal-Wallis test statistics and P-values for the influence of environmental factors on protosteloid abundance and richness. Model = Response × Factor. Abundance refers to scaled abundance per line of substrate. Richness refers to scaled richness per line of substrate. Test statistics are corrected for ties. All models showed significant effects of environmental gradients on scaled abundance and richness.

Click here for additional data file.

Special thanks to John Shadwick and David Orlovich for their help gathering and processing samples, and to the reviewers of this manuscript for many helpful observations and comments.

Additional Information and Declarations

Competing Interests

Author Contributions

The authors declare no competing interests.

Geoffrey Zahn analyzed the data, contributed reagents/materials/analysis tools, wrote the paper, prepared figures and/or tables, reviewed drafts of the paper.

Steven L. Stephenson conceived and designed the experiments, performed the experiments, contributed reagents/materials/analysis tools.

Frederick W. Spiegel conceived and designed the experiments, performed the experiments, contributed reagents/materials/analysis tools, reviewed drafts of the paper.

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
