# Peer review of "Ecological distribution of protosteloid amoebae in New Zealand"

_PeerJ, doi:10.7717/peerj.296_

## Round 0.1 · original submission · Minor Revisions

The revisions you need to make relate mostly to rewriting the manuscript so that it is easier to understand what was done. Please pay particular attention to the reviewer's comments on improving the description of the methodology. In particular, your description of "observations", "lines", "samples" and "sites" could be improved so that readers can understand your sampling sizes and their variation. Was the number of lines the same for all samples? How was abundance determined if you had different numbers of samples per site?

In addition, the discussion focusses mostly on sampling issues and I think it would be more appropriate to emphasise the ecological significance of your results. For instance, you need to discuss the richness and abundance results compared to other studies. How many of the observed species were endemic, native and more widely distributed?

Please clarify your description of latitude. "Increasing latitude" (as you have written in the abstract) and "higher latitudes" (Line 26) means that you are closer to the equator in the southern hemisphere, because latitudes are usually written as negative down here. Perhaps avoid ambiguity by using proximity to the equator or pole as your descriptor of latitude.

Some sections (methods, results, etc.) contain information that belongs in other sections. For example, Lines 79-81 belong in the methods. Also, figure 3 belongs in the results and should be adequately described. Please thoroughly check each section and make sure the content is appropriate to that section.

What does the symbol for Echinostelium bisporum mean in table 1? You should probably include authorities in this table for readers' reference.

Why did you have fewer ground litter samples overall?

Lines 106-109: What does this mean? How were models constructed?

Line 111: "abundance": is this total abundance or did you do this for each species?

Did you test for correlations among your explanatory variables? Did you check for interactions? You need to do these things if they were included in the same GLM. If you used them in separate GLMs you need to make that clear in your description of the modelling methods. It would be nice to see the statistics from your GLM results more clearly presented, such as the coefficient values.

Lines 149-150: What are these percentages compared to to determine what the total number of species present was?

·

Basic reporting

In general the layout of the text is appropriate, in the appropriate order an easy to follow. Some additional information is required in the abstract, in particular to state how samples were analysed to assess diversity and abundance. At the moment it is not clear is this study is based on culture, microscopy or molecular methods until we get to the results. Please indicate how many sites were analysed in this study within the abstract.

Details of how taxa were identified should also be included briefly in the introduction. The validity of using this technique (e.g. as compared to molecular methods) should be touched on in the discussion as it is hard for the reader to understand how hard/easy it is to distinguish between isolates using this approach.

Please note that, in many places, the references are not cited properly. Please make sure you know how to reference. Do not include author initials in your citations and if 3 or more authors, please use the term ‘et al.’.

Please note the term ‘species species’ on line 80.

Where is table 2? Line 84.
Please also note that figure 1 is never actually referred to in the results
Please note the spelling of Auckland (line 115)

I don’t like the presentation of figure 2 and think that all x and y axes should be individually labelled for clarity. Equations for regression linens and correlation coefficients should also be given.

I don’t really know what protosteloid amobae are to so it would be great if you could include a figure that has a few images of some taxa you identified in your samples.

Experimental design

The experimental design is generally fine, and appropriate for this kind of spatial study.
What was the time frame between the collection of sample, air dying and taxonomic assessment? Please make this clear as long periods of sample degradation could impact the validity of your results.

Validity of the findings

How was the effect of different sampling occasions taken into account in your data, and was there any regional bias? For example, were samples taken in the north island taken at one time and those in the south island another? Please make this clear as it impacts the interpretation of latitudinal effects, etc.

Please provide some indication of the likely strength of your microscope/taxonomic key based data. Would you likely have found more diversity in these systems had you adopted a more rigorous molecular approach?

It is quite hard to judge the validity of your findings since information appears to be missing (Table 2).
Also how did you assess the strength of your regressions in a way to determine that they were significant? (line 88)

Reviewer 2 ·

Basic reporting

Portions of the manuscript are unclear or ambiguous.

-The authors' use of the teliological term "preference" should be omitted (Abstract & Lines 81, 84). There is no way to know what protosteloid amoebae prefer or do not prefer. A particular species' occurrence in ground litter, for example, does not necessarily mean it "prefers" to be there.

-Since a large part of this research involved investigations of species distributions "along gradients of climate, elevation, and latitude", more background information would be beneficial defining the ranges of these gradients within the context of the current study. What aspects of climate were specifically investigated, and within each what are the normal limits found within the study areas investigated? Were the ranges "normal" for New Zealand or were these abnormally wet/dry/hot/cold years? Were issues of seasonality controlled for?

-Species "diversity" is used in several places (abstract, line 97), although there is no indication of a specific index or measure of diversity. The authors likely mean to use the term species "richness" throughout.

-Some clarifications and small grammatical corrections are needed:
+Final sentence of the abstract -- unclear whether specifically increasing precipitation and elevation are tied to decrease in species richness, or just a general correlation exists in these two variables.
+Line 15 little in the manuscipt relates directly to the "ecology" of these organisms per se, but rather to distributions, biogeography, and/or correlations with environmental parameters.
+Line 22 New Zealand comprizes many ecosystems, not just one unique one.
+Line 24 and...it...is...
+Line 29 one might argue that the range of latitudes is not necessarily that "wide" compared to, say, Africa or North America.
+Line 35 observations of what?
+Line 48 microscope?
+Line 53 specify which specific climate data
+Line 69 It would be beneficial to remain consistent with terminology - was P. mycophaga "abundant" or "common"? ..."most commonly encountered" suggests it might have been "abundant", but the term "commonly" is misleading, suggesting in occurred in only 1-5% of samples.
+Line 70 Eighty-one out of eighty-two STUDY sites?
+Line 80 species species
+Line 84 Table 2 not found
+Line 87 community richness does not appear to have been measured in this study, perhaps the authors mean "species richness" or Protosteloid amoebae richness".
+Line 89 a definition of "community measures" would be appropriate earlier in the manuscript (Methods).
+Line 96 a "climate" gradient is not recognized from the manuscript, perhaps a "precipitation" gradient is meant?
+Line 102 how many collections comprised the "first collections"?
+Line 104 ...and how many collections comprised the "subsequent collections"?
+Line 103 It would benefit to define "substrate type" as either aerial or ground litter.
+Line 106 to find "substrate species"? Do the author's mean sampling sites?
+Line 120 "...had no impact on the significance of results"...the authors should likey specify the "abundance and richness results".
+LInes 121-122 This sentence is confusing, particularly when P. mycophaga was so abundant, and because no samples were found in supplimentary table 1 that show 486 lines (the highest was found to be 443 at Peel Forest).

-Lines 99-101 Is there a figure that can be used to demonstrate this conclusion? Perhaps refer to Figure 2?

-Figure 1 (Map of sampling locations) would benefit with some labels indicating which island is North, South, Stewart, and Auckland. Also, a map or maps indicating climatic features (elevation and precipitation) would be beneficial.
-Table 1 (Species Observations) would benefit from some explanation of the numbers in the Aerial and Ground columns.
-Figure 2 would benefit from an indication of whether correlations were statistically significant or not.

Experimental design

-A more thorough explanation of the plating of substrates into "lines" is needed to help the reader better distinguish the relative sampling effort between samples of different numbers of "lines". Was one "line" present in one petri plate? Were multiple lines able to be plated within a single plate? Did lines vary in length? How wide were lines of subsrate? Was this method standardized between gound and aerial? Methods described in Spiegel et al. do not specifically discuss the use of these "lines".

-LIne 56 define here what is meant by "microhabitat" (aerial vs ground?)

-The manuscript would benefit if the authors provided a bit more background and discussion of the variety of habitats found throughout the Study area (mixed dry forest, rainforest, podocarp/beech, etc), the relative sampling efforts attempted in those areas, and whether or not particular species or assemblages were discovered to occur in those "habitats".

-Supplimentary Table 1 does not show any sample or site where 486 "lines" were measured. The "Peel Forest" shows 443. This makes the discussion on Lines 111-122 somewhat confusing.

Validity of the findings

The authors would benefit by reiterating that the conclusions are limited to presence/absence data due to differences in sampling effort at different study sites/sampling areas.

+Line 76 Were the previous studies with which the current results are consistent carried out over the same area? Using comparable numbers of collections?

Additional comments

A nice contribution to the current literature on protosteloid amoabae distributrion & occurrence. Some of the writing requires more detail or should be cleaned up (as indicated in comments elsewhere) to make the manuscript more solid.

---

## Round 0.2 · Minor Revisions

I have just a few small minor suggested changes to the text (in a word doc with tracked changes) to improve the readability and a couple of points that I would like you to clarify. I will email you the document shortly.

---

## Round 0.3 · accepted · Accept

Thank you for your quick response to my suggested changes. The only other issue may be the quality of your figures, which look a bit fuzzy. If you need to improve them, I think that the PeerJ staff will be in touch.